# New Insights into Methyl Jasmonate Regulation of Triterpenoid Biosynthesis in Medicinal Fungal Species *Sanghuangporus*
*baumii* (Pilát) L.W. Zhou & Y.C. Dai

**DOI:** 10.3390/jof8090889

**Published:** 2022-08-23

**Authors:** Zengcai Liu, Ruipeng Liu, Xinyu Tong, Li Zou

**Affiliations:** College of Forestry, Northeast Forestry University, Harbin 150040, China

**Keywords:** *Sanghuangporus* *baumii*, methyl jasmonate, terpenoids, triterpenoid biosynthesis, growth and development

## Abstract

Triterpenoids are secondary metabolites produced by the fungus *Sanghuangporus* *baumii* that have important pharmacological activities. However, the yield of triterpenoids is low and cannot meet market demand. Here, we treated *S. baumii* with several concentrations of MeJA (methyl jasmonate) and found that the total triterpenoid content was highest (23.31 mg/g) when the MeJA concentration was 250 μmol/L. qRT-PCR was used to quantify the transcription of five key genes involved in triterpenoid biosynthesis. The results showed that the relative transcription of most genes increased with increasing MeJA concentration, indicating that MeJA is a potent inducer of triterpenoid biosynthesis in *S. baumii*. To further explore whether other terpenoid biosynthesis pathways are also involved in the accumulation of triterpenoids induced by MeJA, we measured the contents of cis-Zeatin (cZ), gibberellins (GAs), and the transcript levels of related biosynthesis genes. We found that MeJA significantly inhibited the biosynthesis of cZ, GAs, and the transcription of related genes. The repressive effects of MeJA on cZ and GA accumulation were further confirmed by growth rate and biomass assays. In conclusion, our study provides an effective method to enhance the triterpenoid content of *S. baumii*, and also provides novel insights into the mechanism of MeJA-induced triterpenoid biosynthesis.

## 1. Introduction

*Sanghuangporus* is an important genus in the Basidiomycota that has a long history of medicinal use in China, where it is known as “forest gold”. *Sanghuangporus* Sheng H. Wu et al. was recently segregated from the broad generic concept of *Inonotus* P. Karst. This genus accommodates some important medicinal fungal species generally called “Sanghuang” in China and Korea, and “Meshimakobu” in Japan [1]. *Sanghuangporus* is mainly used for antibacterial [2], antidiabetic [3], antitumor [4] purposes, and to prolong life [5]. Therefore, *Sanghuangporus* plays a valuable role in human health due to its pharmacological activities. *Sanghuangporus*
*baumii* (Pilát) L.W. Zhou & Y.C. Dai is a parasite on the *Syringa reticulata*. Studies have found that *S. baumii* produces a variety of effective active medicinal ingredients [6,7,8]. Triterpenoids are responsible for the antitumor effect and antioxidant activity of *S. baumii*, and the content of triterpenoids is also an important indicator of the medicinal efficacy and value of *S. baumii* [9].

Because they are secondary metabolites, the content of triterpenoids is low under natural conditions, so various methods must be used to increase the yield of triterpenoids. At present, the common methods used to increase the content of triterpenoids include physical mutagenesis [10], chemical induction [11], and transgenic technology [12], among which chemical induction is the most commonly used and is the most convenient method. Methyl jasmonate (MeJA) is often used as an exogenous chemical inducer and is an important signal transduction molecule in secondary metabolism [13]. The increase in the yield of total triterpenoids by MeJA treatment has been tried and tested, and it has been confirmed in the fungi *Ganoderma lucidum* [14], *Inonotus obliquus* [15], and *Poria cocos* [16], and in the dicot *Chenopodium quinoa* [17]. In the study on *P. cocos*, it was found that adding MeJA can promote transcription of the triterpenoid biosynthesis gene *SQS* and increase the accumulation of total triterpenoids [16]. Similarly, it was also found that MeJA induction could significantly increase the transcription of *HMGR* and *LS*, two genes involved in triterpenoid biosynthesis in *G. lucidum*, and at the same time upregulate the content of total triterpenoids [18]. Previous studies indicated that MeJA can act on CGTCA-motif elements in the promoter regions to activate the expression of key genes in the triterpenoid biosynthesis pathway (mevalonate, MVA), and ultimately promote the accumulation of triterpenoids [14,15,16,17,18]. However, the MVA pathway not only synthesizes triterpenoids, but also synthesizes terpenoids such as cis-Zeatin (cZ) and gibberellins (GAs; Figure 1).Whether the addition of MeJA causes an increase in the triterpenoid content that is related to changes in the expression of other biosynthetic pathway genes remains unknown.

In this study, our objective was to investigate the effect of MeJA treatment on the content of total triterpenoids in *S. baumii*. We used qRT-PCR assays to detect changes in the transcript levels of key genes in the triterpenoid biosynthesis pathway [19,20]; we cloned and sequenced the promoter regions of these key genes to detect the cis-acting elements in their promoter regions. The results will be used to verify the results of previously published studies [15,16]. At the same time, we measured the contents of other terpenoids and the transcript levels of relevant key genes, and the promoter elements of these genes were also analyzed to explore their potential novel regulatory mechanisms to provide a reference for a more comprehensive understanding of the regulation of triterpenoid biosynthesis in *S. baumii*. 

## 2. Materials and Methods

### 2.1. Fungal Strain and Plasmid Vector

*S. baumii* strain DL101 was preserved in the protection laboratory of Northeast Forestry University. It was identified as *S. baumii* based on an ITS sequence alignment (NCBI GenBank No. KP974834). The *Escherichia coli* strain DH5α and the pMD18-T vector were purchased from Takara Biotechnology (Takara, Dalian, China).

### 2.2. MeJA Treatment of Cultures at Different Concentrations

MeJA was dissolved in (Tween 20) and sterilized by filtration through a 0.45 mm Millipore filter before it was added to the PDA (potato dextrose agar) medium. The final concentrations of MeJA were 100, 150, 200, 250, 300, and 350 μmol/L, and control cultures received equal volumes of Tween 20 and sterile water. Cakes of *S. baumii* (1 cm in diameter) were placed in glass Petri dishes containing PDA medium supplemented with different concentrations of MeJA, cultured at 25 °C in the dark, and the mycelial growth rate was measured after 10 d. The mycelial growth rate was calculated by averaging the vertical and horizontal colony diameter measurements. 

The *S. baumii* cakes were inoculated into PD medium to prepare the seed culture. The seed culture was dispersed with a homogenizer and inoculated into 250 mL PD medium at a 4% (*v/v*) inoculum volume and cultured at 25 °C with shaking at 180 rpm for eight days. The PD medium was then supplemented with the same concentration of MeJA as before, and the induction was continued for 48 h before the mycelia were harvested. The *S. baumii* mycelia from the different MeJA treatments were collected and separated into three parts: one part of each sample was frozen at −80 °C for use in qRT-PCR assays and terpenoid hormones measurements; a second part was used for biomass calculation; and the third part was dried at 45 °C for determination of the total triterpenoid content.

### 2.3. Quantification of Terpenoids

The process used to extract the total triterpenoids and the drawing of the standard curve followed previously described methods [8]. Levels of cZ were determined using ELISA kits (mlbio, Shanghai, China) following the manufacturer’s directions. We used a previously described method to measure the GA content and construct the standard curve [21]. Briefly, 0.5 g of each freeze-dried sample was crushed, extracted with 4.5 mL of 70% ethanol, and centrifuged at 5000× *g* for 10 min. The supernatant was then carefully filtered through a 0.45 micron membrane, and 0.5 mL of the filtered sample solution was used in the subsequent reaction. The reaction steps were the same as previously described. The absorbance of the reaction solution was measured in a spectrophotometer at 412 nm, and the GA content was then calculated.

### 2.4. Measurement of Gene Transcript Levels by qRT-PCR

The *S. baumii* mycelia treated with different concentrations of MeJA were collected and total RNA was extracted from each and reverse-transcribed into first-strand cDNA using a Prime ScriptRT reagent Kit (Takara). The cDNAs were then used as the template in qRT-PCR assays. Gene-specific primers were used to evaluate the relative expression of each gene, and the *α*-*tubulin* gene was amplified as the internal control (Appendix A). qRT-PCR reactions were performed in a CFX 96 real-time PCR detection system (Bio-Rad, Hercules, CA, USA), and the relative gene expression was calculated using the 2^−ΔΔCt^ method of Livak and Schmittgen [22]. Mycelia treated with sterile water were defined as the control samples, and the expression of genes in the H_2_O-treated mycelia were set to 1.0 to normalize the data for the different MeJA treatments.

### 2.5. Amplification the Gene Promoter Regions

DNA was extracted from *S. baumii* using the CTAB method [23] and was then used as the template for amplification and cloning of the promoter regions. PCR amplification was performed using *S. baumii* genomic DNA as the template and a pair of forward and reverse primers specific to the promoter regions (Appendix A). The PCR products were purified and cloned into the pMD18-T vector, and then transformed into *E. coli* DH5α. The positive clones were identified by amplification with primers pMD18-F and pMD18-R, and the inserts were then sequenced (TsingKe, Beijing, China). The sequences were analyzed with PlantCARE (http://bioinformatics.psb.ugent.be/webtools/plantcare/html/ (accessed on 20 May 2022) to predict the functional cis-acting elements in the promoter regions.

### 2.6. Statistical Analysis

All experiments were performed in triplicate, and the standard deviations (±SD) are indicated by error bars in the figures. The Duncan test and Spearman method as implemented in SPSS 17.0 were used for significance analysis and correlation analysis, respectively. *p* < 0.05 was considered to be a significant difference as compared to the control, and *p* < 0.01 was considered to be an extremely significant difference.

## 3. Results

### 3.1. Total Triterpenoid Profiles in Response to MeJA Treatment in S. baumii

To detect the effect of MeJA treatment on the total triterpenoid content in *S. baumii*, the total triterpenoid contents in the MeJA treatments at different concentrations were determined. The results (Figure 1) showed that as the MeJA concentration increased from 100 to 250 μmol/L, the total triterpenoid contents increased significantly. The total triterpenoid concentration peaked at 250 μmol/L MeJA, and was 1.62 times that of the control (H_2_O), indicating that MeJA induced the most significant increase in total triterpenoid synthesis at this concentration. At MeJA concentrations >250 μmol/L, the content of total triterpenoids decreased significantly; the highest concentration of MeJA (350 μmol/L) had an inhibitory effect on total triterpenoid production, and it decreased to a level significantly lower than that of the control. The total triterpenoid content in the Tween 20 control treatment was slightly higher than that in the H_2_O control group, suggesting that Tween 20 may have a positive effect on the synthesis of total triterpenoids in *S. baumii*.

### 3.2. Relative Changes in the Expression of Five Genes in Response to MeJA Treatment in S. baumii

Treatment with MeJA was found to significantly increase the content of total triterpenoids, most likely by increasing the expression of key genes in the triterpenoid biosynthesis pathway. We examined the changes in the transcript levels of five key triterpenoid biosynthesis pathway genes. It can be seen from the results that the expression of four genes showed a general upward trend with increasing concentrations of MeJA, indicating that MeJA can promote the transcription of these genes (Figure 2a–d). In this experiment, the transcript level of the *LS* gene showed a downward trend (Figure 2e), indicating that MeJA treatment may inhibit its expression.

To further study the correlations between MeJA concentration, total triterpenoid contents, and the transcript levels of five triterpenoid biosynthesis genes, we performed a Spearman correlation analysis (Appendix A). The results showed that the concentration of MeJA was not significantly positively correlated with the total triterpenoid content, indicating that increases in MeJA concentration could not cause a continuous increase in the total triterpenoid content in *S. baumii*. The results of the analysis of total triterpenoid content and the transcript levels of the triterpenoid biosynthesis genes showed that the relative transcription of the *HMGR*, *IDI*, *SQS*, and *SE* genes was significantly positively correlated with total triterpenoid content, suggesting that these genes play an important role in triterpenoid biosynthesis.

### 3.3. Analysis of the Promoter Regions of Five Genes Involved in Triterpenoid Biosynthesis

Gene promoter sequences usually contain multiple functional elements that play roles in the response to external stresses and transcriptional induction [24]. To determine whether the expression of the five triterpenoid biosynthesis genes respond to MeJA treatment, we cloned and analyzed their promoter regions. Among them, the *SQS* and *LS* gene promoters have been shown to contain MeJA-responsive elements [25,26]. We also identified MeJA-responsive elements in the promoters of the other three genes (Figure 3a–c). The *HMGR* gene promoter contains typical conserved cis-acting elements such as the CAAT-box (controls the frequency of transcription initiation) and the TATA-box (determines the initiation site of RNA synthesis), as well as a MeJA responsive element (the CGGTA-motif), light responsive elements, and low temperature responsive elements (Figure 3a). Similarly, the *IDI* and *SE* gene promoters also include CAAT-boxes, TATA-boxes, and the CGGTA-motif. However, the *IDI* gene promoter also includes an auxin-responsive element (TGA-element) and a low temperature responsive element, in addition to others (Figure 3b). The *SE* gene promoter includes abscisic acid responsive elements (ABRE) and defense and stress responsive elements (TC-rich repeats), (Figure 3c), suggesting that these genes have the ability to respond to multiple exogenous signals.

### 3.4. cZ Profile and TRIT1 Gene Expression Changes in Response to MeJA Treatment in S. baumii

In addition to triterpenoid synthesis, the MVA pathway can also synthesize various terpenoids such as cZ and GAs (Figure 1). Therefore, the contents of cZ and GAs, and the transcript levels of related genes were analyzed. The results showed that the cZ content in *S. baumii* decreased significantly after MeJA induction, and that the cZ content was the lowest (0.99 mg/g) in the 250 μmol/L MeJA treatment; 0.31 mg/g lower than the H_2_O control (Figure 1). Furthermore, the transcript level of the cZ synthesis gene *TRIT1* decreased significantly in response to MeJA treatment (Figure 4a), and the inhibitory effect on *TRIT1* transcription was more significant at MeJA concentrations >100 μmol/L. The *TRIT1* gene transcript level was the lowest in the 350 μmol/L MeJA treatment, and was only 20% that of the control (H_2_O). 

The correlations between MeJA concentration, *TRIT1* gene transcript level, and cZ content were further analyzed. The results showed that MeJA concentration was significantly negatively correlated with the *TRIT1* gene transcript level and cZ content (Appendix A), indicating that MeJA repressed *TRIT1* gene transcription and the accumulation of cZ in a concentration-dependent manner. The changes in *TRIT1* expression and cZ content showed a significant positive correlation, indicating that the *TRIT1* gene is involved in and positively regulates the synthesis of cZ.

### 3.5. GAs Profile and GGPS Gene Transcription Changes in Response to MeJA Treatment in S. baumii

The GA content in *S. baumii* treated with different concentrations of MeJA was determined. The results showed that the content of Gas was reduced by MeJA treatment (Figure 1). The GA content in the 250 μmol/L MeJA treatment was the lowest (0.33 mg/g), and was only about 40% the content in the control (H_2_O). Furthermore, the transcript level of the *GGPS* gene was significantly reduced after MeJA treatment, and the inhibitory effect on the transcription of *GGPS* was more significant when the concentration of MeJA was >100 μmol/L. The lowest level of *GGPS* transcription was in the 350 μmol/L MeJA treatment, and it was only around 25% that of the H_2_O control (Figure 4b).

In order to examine the correlations between MeJA concentration, *GGPS* gene transcript level, and GAs content, we performed a Spearman correlation analysis. The results showed that the MeJA concentration was significantly negatively correlated with *GGPS* gene transcript level and GA content (Appendix A), indicating that MeJA treatment had a concentration-dependent effect on *GGPS* gene expression and GA accumulation. There was a significant positive correlation between *GGPS* gene expression and GA content, indicating that the *GGPS* gene is involved in and positively regulates the synthesis of GAs in *S. baumii*.

### 3.6. Analysis of the Promoter Regions of the TRIT1 and GGPS Genes

The functional elements present in the *TRIT1* gene promoter region were predicted using PlantCARE (Figure 3d). The results showed that the promoter region contains the typical conserved CAAT-box and TATA-box elements, and also has a variety of hormone-responsive elements, including MeJA-responsive elements, GA-responsive elements (P-box, GARE-motif), abscisic acid-responsive elements, auxin-responsive elements, and defense- and pressure-responsive elements. We also used PLantCARE to predict the functional elements in the *GGPS* gene promoter region. As shown in Figure 3e, the promoter region contains the typical conserved CAAT-box and TATA-box elements, as well as the MeJA-responsive element, the anaerobic inducible element (ARE), and the drought inducible element (MBS). MeJA-responsive elements were found in the promoter regions of both *TRIT1* and *GGPS*, suggesting that these genes may also respond to MeJA, but the effect may be repressive. 

### 3.7. Effects of MeJA on Mycelial Growth in S. baumii

GAs are important growth hormones that act to promote the elongation and growth of plants [27], and also significantly promote the growth and development of bacteria and fungi [28]. Cytokinins also regulate growth and development, and play roles in delaying senescence, promoting cell division, and stimulating the growth of lateral buds [29]. We speculated that these two terpenoid hormones may show a correlation with the growth of *S. baumii* mycelia. The GA and cZ contents decreased significantly in response to MeJA treatment (Figure 1), so we measured the mycelia growth rate and biomass after MeJA treatment. The results showed that the phenotypes of the *S. baumii* cultures were quite different when grown under different concentrations of MeJA. The colonies were white after treatment with H_2_O and Tween 20. However, the yellow centers of the colonies expanded considerably when grown on media containing MeJA (Figure 5a). In addition, the mycelial growth rates and colony diameters of *S. baumii* were significantly affected by the different concentrations of MeJA. The growth rate of the mycelia was inhibited in a concentration dependent manner at MeJA concentrations >100 μmol/L. The mycelial growth rate was the fastest (0.4 cm/d) and the biomass was the highest (4.39 g/L), in the H_2_O control, and this was significantly different from the other treatment groups. After treatment with 350 μmol/L MeJA, the growth rate of mycelia was the slowest (0.1 cm/d) and the biomass was the lowest (1.87 g/L; Figure 5b,c), indicating that MeJA treatment inhibits the growth of *S. baumii* mycelia.

In order to study the effect of MeJA induction on mycelial growth rate and biomass, we performed a Spearman correlation analysis. The results showed that MeJA concentration was significantly negatively correlated with mycelial growth rate and biomass (Appendix A), indicating that these related traits were significantly inhibited by increasing MeJA concentrations. Spearman correlation analysis was used to examine the relationships between mycelial growth rate, biomass, and the contents of cZ and GAs in the MeJA treatments (Appendix A). We identified a significant positive correlation between the levels of the two terpenoid hormones (cZ and GAs) and the growth of *S. baumii*. This indicated that cZ and GAs may be involved in and positively regulating the growth and development of *S. baumii*.

## 4. Discussion

MeJA has a broad spectrum of physiological effects; not only does it regulate the growth and development of plants [30], but it also participates in the defense response of plants to biotic and abiotic stresses [31,32] and can induce the accumulation of secondary metabolites in fungi [14,15,16]. In this study, we treated cultures of *S. baumii* with concentrations of MeJA between 100 and 350 μmol/L to determine the effects of MeJA on the biosynthesis of triterpenoids. We found that MeJA treatment significantly increased the content of total triterpenoids within a certain concentration range, and that the expression of most key genes involved in triterpenoid biosynthesis responded positively to MeJA treatment. We confirmed that the accumulation of triterpenoids is closely related to the expression of key genes in the triterpenoid biosynthetic pathways [16,18]. By analyzing the promoter sequences of these genes, we found that they all possess the basic elements for transcriptional activity (CAAT-box and TATA-box). In addition, various light-responsive elements present in the promoters suggest that these genes may be light-inducible [33]. Some stress-responsive elements involved in the defense against environmental perturbations were also identified [34]. The most important thing to note is that all of these gene promoters contain the MeJA responsive element, the CGTCA motif, which is also an important reason why they are closely related to the accumulation of triterpenoids in response to MeJA treatment [25,26]. These results are basically consistent with previous conclusions, “MeJA acts on CGTCA-motif elements of key gene promoter through signal transduction. The promoter activates the expression of key gene involved in the triterpenoid biosynthesis pathway, and ultimately promotes triterpenoid accumulation” [16,18]. MeJA-responsive elements were identified in the promoter regions of the *HMGR*, *IDI*, *SQS*, *SE*, and *LS* genes, but which of these promoters can mediate the response to MeJA will require further study.

On the basis of previous studies on the mechanism of triterpenoid biosynthesis induced by MeJA [18], we continued to explore the changes in gene transcript levels and metabolite contents of other terpenoids from the MVA biosynthesis pathway after MeJA treatment. We found that the content of cZ and the relative transcription of the key gene *TRIT1*, as well as the GA content and the transcription of the key gene *GGPS*, were significantly negatively correlated with the MeJA concentration, indicating that MeJA treatment inhibited the expression of other terpenoid biosynthesis pathway genes and metabolite production. At the same time, we identified MeJA-responsive elements in the promoters of the *TRIT1* and *GGPS* genes. However, MeJA significantly inhibited the transcription of these two terpenoid biosynthesis genes, which was in contrast to the effect of MeJA treatment on the expression of key triterpenoid biosynthesis genes. Furthermore, MeJA significantly downregulated the contents of two terpenoid hormones related to growth and development. The reduction in the levels of cZ and GAs will inevitably retard growth and development [27,29,35]. In this work, *S. baumii* showed markedly inhibited mycelial growth and reduced mycelial biomass when grown on medium containing MeJA, especially at concentrations >100 μmol/L (Figure 5). Surprisingly, a similar phenomenon has been observed in studies involving the overexpression of triterpenoid biosynthesis genes to promote triterpenoid accumulation. The increase in triterpenoids and the decrease in mycelial biomass are likely to be caused by reduced levels of terpenoid hormones [36,37,38], but this needs to be investigated in future work.

## 5. Conclusions

In conclusion, we propose a more comprehensive biosynthesis mechanism for MeJA-induced triterpenoid production in *S. baumii*: MeJA acts on the CGTCA motif, a cis-acting element located in the gene promoters, through signal transduction to activate the transcription of key genes in the triterpenoid biosynthesis pathway, thereby increasing key gene expression and triterpenoid accumulation. In addition, MeJA also acts on inhibiting the activity of CGTCA-motif elements in the promoters of key genes in other terpenoid biosynthesis pathways, reducing the expression of these genes and the accumulation of terpenoids from other biosynthesis pathways. The accumulation of triterpenoids was finally promoted under the conditions of cooperation between the two parties. The experimental results of our study provide a new insight into the mechanism by which MeJA acts to increase triterpenoid yield in *S. baumii*, and also provide a reference for a more comprehensive understanding of the regulation of genes involved in triterpenoid biosynthesis.

## Figures and Tables

**Figure 1 jof-08-00889-f001:**
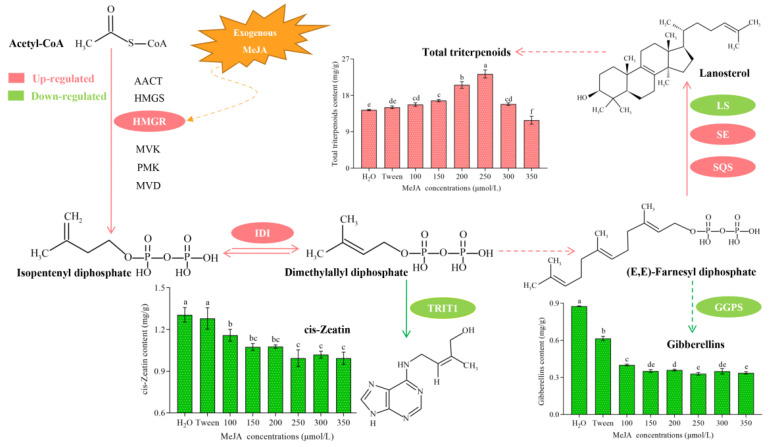
The mevalonate (MVA) biosynthetic pathway and changes in the contents of total triterpenoids and two terpenoid hormones (cZ and GAs) in response to treatment with different concentrations of MeJA. The mean difference being significant at the 0.05 level, *p* < 0.05.

**Figure 2 jof-08-00889-f002:**
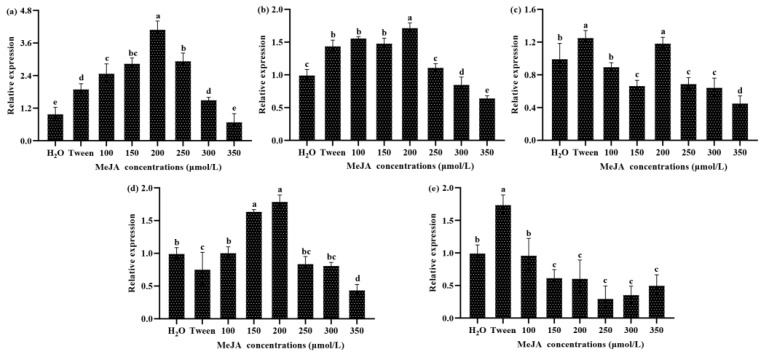
Changes in the transcript levels of five triterpenoid biosynthesis genes in response to MeJA concentrations *in S. baumii*. qRT-PCR assays were used to calculate the relative expression of the (**a**) *HMGR*, (**b**) *IDI*, (**c**) *SQS*, (**d**) *SE*, and (**e**) *LS* genes. The mean difference being significant at the 0.05 level, *p* < 0.05.

**Figure 3 jof-08-00889-f003:**
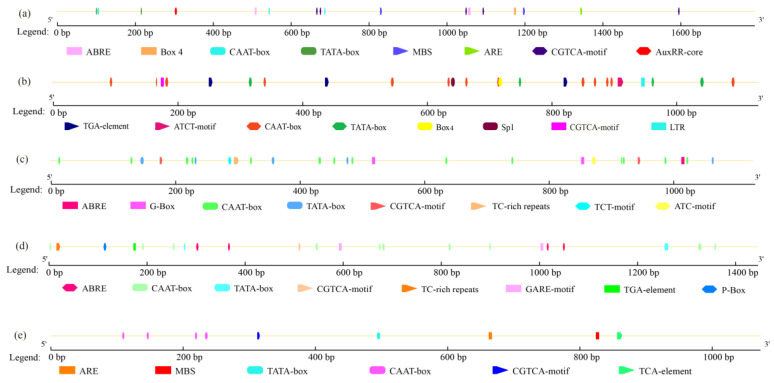
Major cis-acting elements and their positions in the promoter regions of five terpenoid biosynthesis genes. Schematic diagrams of the (**a**) *HMGR* gene promoter; (**b**) *IDI* gene promoter; (**c**) *SE* gene promoter; (**d**) *TRIT1* gene promoter; and (**e**) *GGPS* gene promoter showing the color coded cis-acting elements.

**Figure 4 jof-08-00889-f004:**
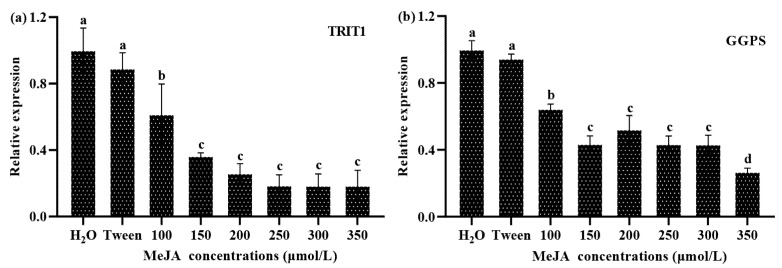
The relative expression of the *TRIT1* (**a**) and *GGPS* (**b**) genes in *S. baumii* treated with MeJA at different concentrations. The two controls treatments were H_2_O and Tween. The mean difference being significant at the 0.05 level, *p* < 0.05.

**Figure 5 jof-08-00889-f005:**
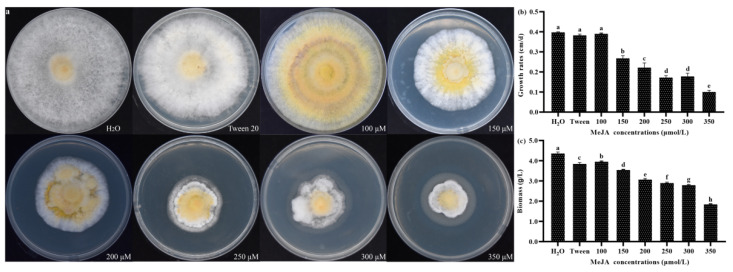
Growth of *S. baumii* colonies on PDA (potato dextrose agar) medium supplemented with different concentrations of MeJA. (**a**) Colony morphology; (**b**) growth rate; (**c**) biomass. The mean difference being significant at the 0.05 level, *p* < 0.05.

## Data Availability

All data is contained within the article.

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
