# Peer review of "New Insights into Methyl Jasmonate Regulation of Triterpenoid Biosynthesis in Medicinal Fungal Species Sanghuangporusbaumii (Pilát) L.W. Zhou & Y.C. Dai"

_jof, 2022, doi:10.3390/jof8090889_

Round 1

Reviewer 1 Report

  Dear authors, 

your work is excellent but  I would some addings to be more easier for wider scientific audience to understand and accept...

           I am attaching the file where i have put some suggestions.....

please indicate the relevance of this research in the conlusion section...

  All the best for further research with other fungal species....

Author Response

Thanks for the comments. All comments are constructive, and revisions have been made based on the suggestions (coloured red in the revised manuscript).

  1. Suggestion – change the title to the following

New insight into methyl jasmonate regulation of triterpenoids biosynthesis in medicinal fungal species Sanghuangporus baumii (Pilát) L.W. Zhou & Y.C. Dai

A: The title has been modified as you suggested. “New insight into methyl jasmonate regulation of triterpenoids biosynthesis in medicinal fungal species Sanghuangporus baumii (Pilát) L.W. Zhou & Y.C. Dai”

  1. Introduction Add the most important data of taxonomy of the fungal Species – more biological facts about the specific species is necessary to improve the application values of this research

e.g.  Sanghuangporus baumii, which grows on Syringa in the wild, is also served as medicinal fungus in China. Sanghuangporus accommodates some important medicinal fungal species generally are called “Sanghuang” (means yellow organism grows on Morus) in China and Korea,  and  “Meshimakobu”  in  Japan.  Sanghuangporus Sheng H. Wu et al. and Tropicoporus L.W. Zhou et al. were recently segregated from the broad generic concept of Inonotus P. Karst (Zhou et al. 2016). The former two genera differ from Inonotus s. str. chiefly in having dimitic hyphal system. Sanghuangporus is characterized by perennial and effused-reflexed to pileate basidiomata, occurring in a variety of climate environment, whereas Tropicoporus is distinguished by annual to perennial basidiomata, and a tropical distribution (Zhou et al. 2016).

A: Some important data of taxonomy of Sanghuangporus has been added to the manuscript based on the suggestion. “Sanghuangporus Sheng H. Wu et al. was recently segregated from the broad generic concept of Inonotus P. Karst. This genus accommodates some important medicinal fungal species generally are called “Sanghuang” in China and Korea, and “Meshimakobu” in Japan.”

3.Please add more recent publications (within the last 5 years) which are relevant for the taxonomy of the species as well as on the medicinal qualities of analyzed species…..suggestion  is given in the comments

https://www.sciencedirect.com/science/article/abs/pii/S1874390019308729

Cyclofarnesane sesquiterpenoids from the fungus Sanghuangporus sp.

A: We have added more recent publications (within the past 5 years) related to the taxonomy of species as well as on the medicinal qualities of analyzed species as suggested in the comments. “Sanghuangporus Sheng H. Wu et al. was recently segregated from the broad generic concept of Inonotus P. Karst. This genus accommodates some important medicinal fungal species generally are called “Sanghuang” in China and Korea, and “Meshimakobu” in Japan [1]. Sanghuangporus is mainly used for anti-bacterial [2], anti-diabetic [3], anti-tumor [4], and to prolong life [5].Triterpenoids are responsible for the anti-tumor effect and anti-oxidant activity of S. baumii, and the content of triterpenoids is also an important indicator of the medicinal efficacy and value of S. baumii [9].”

Reviewer 2 Report

This manuscript reports the use of the plant hormone, methyl jasmonate (MeJA), for the enhancement of triterpenoid production in the fungus Sanghuangporus baumii. The effects of MeJA on the biosynthesis of triterpenoids were carefully investigated. Addition of MeJA was found to inhibit the fungal growth (less biomass obtained), however it stimulated the production of triterpenoids. qRT-PCR revealed changes of transcription of five key genes, which involved in triterpenoids biosynthesis. MeJA significantly inhibited the biosynthesis of cis-Zeatin (cZ) and gibberellins (GAs). Comments and suggestions are listed below.

The weakness of this work is that it is not known which triterpenes are induced after the addition of MeJA. The mevalonate (MVA) biosynthetic pathway also involves in the biosynthesis of common compounds, i.e. cholesterol or ergosterol, and these common metabolites will also be detected in the “total triterpenoids”. Further work is needed to identify the triterpenoid that was increased after the addition of MeJA. However, this is only a suggestion, not required for revision.    

Other comments and suggestions:

1.     Abstract; “Here, we treated S. baumii with several concentrations of MeJA and…”; please provide a full name of “MeJA”.

2.     Keywords should include the word “Triterpenoid biosynthesis”.

3.     Please revise “In conclusion, our study provides an effective method for increasing the triterpenoids content of S. baumii, and…” to “In conclusion, our study provides an effective method to enhance the triterpenoid content of S. baumii, and…”.

4.     “However, the MVA pathway not only synthesizes triterpenoids…”; a full name for “MVA” is important before using the abbreviation.

5.     Compounds in the genus Sanghuangporus have interesting antioxidant activity, including triterpenoids. So, it is worth mentioning this rationale why this work investigated the compounds in this fungus, please see A rapid analysis of antioxidants in Sanghuangporus baumii by online extraction-HPLC-ABTS, RSC Adv., 2021,11, 25646-25652; Extraction and antioxidant activity of total triterpenoids in the mycelium of a medicinal fungus, Sanghuangporus sanghuang. Sci Rep 9, 7418 (2019). https://doi.org/10.1038/s41598-019-43886-0

6.     Figure 2; pictures of graphs are not clear. Please improve.

Author Response

Thanks very much for the suggestion. We have altered this in the revised manuscript.

  1. Abstract; “Here, we treated S. baumii with several concentrations of MeJA and…”; please provide a full name of “MeJA”.

A: The full name of “MeJA” has been provided. (methyl jasmonate)

  1. Keywords should include the word “Triterpenoid biosynthesis”.

A: “Triterpenoids biosynthesis” has been added to keywords.

  1. Please revise “In conclusion, our study provides an effective method for increasing the triterpenoids content of S. baumii, and…” to “In conclusion, our study provides an effective method to enhance the triterpenoid content of S. baumii, and…”.

A: This sentence has been modified according to the comments. “In conclusion, our study provides an effective method to enhance the triterpenoids content of S. baumii

  1. “However, the MVA pathway not only synthesizes triterpenoids…”; a full name for “MVA” is important before using the abbreviation.

A: The full name of “MVA” has been provided. (mevalonate)

  1. Compounds in the genus Sanghuangporus have interesting antioxidant activity, including triterpenoids. So, it is worth mentioning this rationale why this work investigated the compounds in this fungus, please see A rapid analysis of antioxidants in Sanghuangporus baumii by online extraction-HPLC-ABTS, RSC Adv., 2021,11, 25646-25652; Extraction and antioxidant activity of total triterpenoids in the mycelium of a medicinal fungus, Sanghuangporus sanghuang. Sci Rep 9, 7418 (2019). https://doi.org/10.1038/s41598-019-43886-0

A: We have revised this section according to your suggestion. “Triterpenoids are responsible for the anti-tumor effect and anti-oxidant activity of S. baumii, and the content of triterpenoids is also an important indicator of the medicinal efficacy and value of S. baumii [9].”

  1. Figure 2; pictures of graphs are not clear. Please improve.

A: We have uploaded high quality pictures in the system.